# The Effect of Acidic and Alkaline pH on the Physico-Mechanical Properties of Surimi-Based Edible Films Incorporated with Green Tea Extract

**DOI:** 10.3390/polym12102281

**Published:** 2020-10-05

**Authors:** Sadia Munir, Miral Javed, Yang Hu, Youming Liu, Shanbai Xiong

**Affiliations:** 1College of Food Science & Technology, Huazhong Agricultural University, Wuhan 430070, China; bajwa.uos@gmail.com (S.M.); miralfst23@gmail.com (M.J.); huyang@mail.hzau.edu.cn (Y.H.); xiongsb@mail.hzau.edu.cn (S.X.); 2National R & D Branch Center for Conventional Freshwater Fish Processing, Wuhan 430070, China

**Keywords:** surimi, edible films, physico-mechanical, permeability, green tea extract, thermal stability

## Abstract

The effects of green tea extract (GTE) at acidic and alkaline pH (pH 3 and 11, respectively) on the physico-mechanical, thermal, and water transmission properties of silver carp surimi-based edible films were investigated. Incorporation of GTE significantly (*p* < 0.05) reduced elongation at break (EAB) but improved tensile strength (TS). Significant (*p* < 0.05) decreases in the solubility of films and water vapor permeability (WVP) were also perceived in GTE-containing films. Transparency and color were also affected, depending upon the concentration and pH. Films prepared at an acidic pH demonstrated significantly (*p* < 0.05) improved WVP, transparency, solubility, and thermal stability compared to those prepared at an alkaline pH. The protein pattern of films revealed a strong interaction between phenolic compounds of GTE and surimi proteins as evidenced by the presence of a myosin heavy chain (MCH) at the top of the gel. Generally, the addition of GTE at an acidic pH had significantly (*p* < 0.05) greater impact on film properties than at an alkaline pH and could offer great potential for surimi protein-based edible films with improved properties.

## 1. Introduction

An increasing interest among consumers regarding health has stimulated researchers to focus on novel techniques for extending the shelf life of food commodities without using additives and preservatives [1]. Environmental concerns were also basic motivations for the formulation of bio-based or biodegradable films from renewable natural polymer-based constituents, such as polysaccharides, lipids, and proteins. Among these, films based on protein have received massive attention due to superior biodegradability, bioavailability, water barrier properties, and prospective uses in foodstuff packaging [2,3]. Fish muscle (surimi)-based proteins have been used as a film-forming solution to prepare edible films [3]. Fish myofibrillar protein (FMP), the main constituent of surimi, has shown outstanding film-forming capabilities in acidic and alkaline conditions [4], better than other well-known proteins [3,5,6]. Silver carp is one of the most commonly cultured fish in China, with an annual production of around 3.86 million tons in 2018 [7]. Therefore, the formulation of active packaging materials from silver carp muscle protein is an appealing alternative.

The use of plant-based natural active agents has received enormous attention in terms of boosting the mechanical and barrier properties, glass transition temperature [2,3,5], and antioxidant [3,8] and antimicrobial properties of protein-based films [5,9] and to reduce the water vapor transmission [6]. Hydroxyl groups of plant-extract phenolic compounds are capable of interacting with the proteins and thus can modify the structural and functional properties of the proteins. Phenolic compounds can attach to the proteins through hydrogen and covalent bindings. Furthermore, hydrophobic and hydrophilic associations among phenolic compounds and proteins have also been well established [9,10,11]. The phenolic–protein cross-link is generally disturbed by pH, protein sources, process temperature, and the type and structure of the interaction developed during the film-forming procedure.

Green tea extract is a good source of phenolic compounds [12]. Chemically, green tea is a plentiful source of flavonols, flavanols, and theaflavins. The most important and dominant flavanols are catechins, and the common forms of catechins are epicatechin (EC), gallocatechin (GC), catechin gallate (CG), epigallocatechin (EGC), epicatechin gallate (ECG), gallocatechin gallate (GCG), and epigallocatechin gallate (EGCG) [13]. Among all types of catechin derivatives, EGCG was described as the most generous form of catechin, comprising about 50%–80% of total catechins originating in green tea leaves [14]. Phenolic compounds derived from green tea are proven to have antimicrobial and antioxidant properties [13,15,16]. In addition, they are low-cost, nontoxic, tolerable to consumers, and biocompatible, making them a perfect candidate as a potent source of antioxidant, antimicrobial, and anticarcinogenic properties in minimally processed foods [16].

The utilization of pH 11 for the chemical interactions between proteins and phenolic compounds has been studied by different research groups [2,3,10]. On the other hand, the chemical interactions between polyphenols and fish proteins at acidic pH was investigated in [6,17,18]. However, the incorporating effects of green tea extract (GTE) in both acidic (pH 3) and alkaline (pH 11) conditions in comparison to surimi-based edible films have not been studied. The main objective of the study was to investigate the effect that acidic and alkaline pH values had on the properties, including the physico-mechanical, thermal, and barrier properties, of surimi-based edible films incorporated with GTE.

## 2. Material and Methods

### 2.1. Materials

Frozen surimi of silver carp utilized in the present work was purchased from Honghu Jingli Aquatic Food Co. Ltd. (Honghu, China) and deposited at −20 °C for the all experiments. GTE was acquired from Xi’an Reain Biotechnology Co. Ltd. (Xi’an, China). SDS–PAGE chemicals were purchased from Bio-Rad (Hercules, CA, USA). The chemicals used during experiments were of analytical grade.

### 2.2. Preparation of Film-Forming Solution

Film-forming solution was prepared by following the earlier study by Munir et al. [6]. Frozen surimi used in the present study was thawed at 4 °C overnight in a refrigerator and sliced into ∼1 cm × 1 cm × 1 cm cubic pieces. Thereafter, 16 g of thawed surimi was dispersed in 84 mL of distilled water in the presence of 2.5% NaCl (*w*/*w*) and 0.25% Tween 80 (*w*/*v*, based on the solution) in a glass beaker. Consequently, the mixture was homogenized using a homogenizer (IKA Labortechnik, Selangor, Malaysia) at 13,000 rpm for 90 s. The protein concentration was determined using the Lowery method, and then glycerol was also added to the solution on the base of 60% (*w*/*w*) of protein. Then, the obtained mixture was stirred at <10 °C for 50 min. The pH of the resulting mixture was subsequently adjusted to acidic (pH 3) or alkaline (pH 11) conditions by 1 mol/L HCl or 1 mol/L NaOH.

Green tea extract (GTE) was dissolved in the mixture at 0, 2%, 4%, and 6% (*w*/*w*, based on protein), and the resulting blend was again agitated for 1 h at 20–25 °C, and the pH of the blend was readjusted.

The air bubbles of the blend were removed by using a centrifuge (Beckman Coulter, CA, USA) for 10 min at 3500 rpm min. After the removal of air bubbles, the final mixture was filtered carefully through cheesecloth to eliminate insoluble residues. Consequently, 50 mL of the mixture was poured on a 10 cm× 15 cm mold or frame, and then a hot air dryer (Jianghong experimental equipment CO, Ltd, Shanghai, China) (37 °C for 12 h) was used for drying. After that, films were manually peeled off by hand and conditioned for 4 days in a conservational chamber to attain the corresponding moisture constituents (55–60%) of the final films for further analysis.

### 2.3. Determination of Film Properties

#### 2.3.1. Measurement of Film Thickness

Film thickness was determined according to our previous study [6]. Six distinctive places on each film sample were measured to check the thickness using a mechanical thickness gauge (HY-699E, Knicks Testing Technology Co., Ltd., Tianjin, China).

#### 2.3.2. Film Solubility

Measurement of the solubility of surimi films was implemented by following the method of Gennadios et al. [19] with small alterations. A proportion (4 cm× 10 cm) of pre-dried surimi films containing GTE was weighted and submerged in tubes containing 20 mL of distilled water. The blend in tubes was then placed in a shaker (shaking at 150 rpm, at 30 °C for 24 h) (Heidolph Incubator 1000, Heidolph Instruments, Schwabach, Germany). The undissolved proportion of films was collected and dried at 105 °C for 24 h. WS was calculated as follows:
Film solubility (%) = (Initial dry weight (*W_₀_*) − final dry weight (W_f_)/initial dry weight (*W_₀_*) × 100(1)

Six trials were examined for each film.

#### 2.3.3. Light Transparency of Films

Analyses of the light transparency of the edible films were performed at the selected range of wavelength (200–800 nm) following the method of Shiku et al. [20]. A UV–visible spectrophotometer (model UV-1800, Shimadzu, Japan) was used to determine the light transparency of films. All trials were repeated six times. The transparency proportion of the films was calculated using the following equation:
Transparency = −log T_600_/x(2)
where T_600_ symbolizes (%) transfer of transparency at a range of 600 nm, and x denotes the thickness (mm) of films. A higher light transparency assessment means a lower transfer of light around films.

#### 2.3.4. Water Vapor Permeability (WVP)

The permeability of water across the surimi films was conducted in triplicate according to the method of Shiku et al. [20] with slight adjustments. The film was sealed on a glass permeation cup containing silica gel (0% RH) by using silicone vacuum grease and a rubber band to hold the film. The cups were placed in a desiccator with distilled water at 30 °C. The cups were weighed at 1 h intervals over a period of 8 h. WVP of the film was calculated as follows:
WVP = w × x/A × t × (P_2_ − P_1_),(3)
where w signifies the gain in weight (g), x denotes film thickness (m), A denotes the visible area of the film (m^2^), t is the time of weight gain (s), and P_2_ − P_1_ is the vapor pressure difference across the film (Pa).

#### 2.3.5. Mechanical Properties

For the measurement of tensile strength (TS) and percent elongation at break (EAB), an electronic texture analyzer Model TA. XT Plus (Stable Micro System Goldaming, Goldaming, UK) was used as specified by Iwata et al. [21] with slight modifications. The film samples with a length of 80 mm, a width of 40 mm, and thickness of 0.087–0.107 mm was used to calculate EAB and TS and then were fixed between the extension grips of the instrument. The initial gauge separation and test speed were, respectively, 35 mm and 5 mm/s. TS (MPa) was calculated as follows:
TS = F_max_/Φ(4)
where F_max_ represents the maximum load (N) to pull the sample apart, and Φ denotes the cross-sectional area.
EAB (%) = Δl/Ɩ₀ × 100(5)

EAB was calculated as a film extension ratio, where Δl represents the film extension at the point of sample rupture, and l₀ is the initial length of the film sample.

#### 2.3.6. Determination of Color Difference

The color of edible films was detected using a CIE colorimeter (Hunter Associates Laboratory, Inc., Reston, VA, USA), lab system where the L*, a*, b*, parameters represent, respectively, lightness, redness/greenness and yellowness/blueness of the surimi films and a white plate was used as a standard reference (L* = 95.23, a* = −0.59, b* = 3.31). Color was expressed as L*, a*, b*, and (ΔE*) values. Total difference in color (ΔE*) was calculated as described by Gennadios et al. [22] in the following equation:(6)$$\Delta E*=\sqrt[{}]{{(\Delta L*)}^{2}+(\Delta {a*)}^{2}+(\Delta {b*)}^{2}}$$
where ΔL*, Δa*, and Δb* are the differences between the related color parameters of the samples.

#### 2.3.7. Thermal Analysis of Films

Thermal analysis of films was performed using a thermo-gravimetric analyzer (TGA) following the protocol described by Hoque et al. [23]. A Hi-Res TGA 2950 thermo-gravimetric analyzer (TA Instrument, New Castle, DE, USA) was used for the thermal study. Surimi-based film samples (5 mg) were scanned from 30 °C to 600 °C at a rate of 10 °C/min, under a nitrogen-rich atmosphere at a flow rate of 10 mL/min.

#### 2.3.8. Protein Pattern Analysis of Films

Protein pattern analysis of surimi-based films was performed according to the method described by Nie et al. [2]. The samples (0.1 g) were processed to determine the protein pattern of GTE-incorporated films with acidic and alkaline pH, respectively. Then, 10 µL of each prepared film sample was loaded onto 5% and 10% polyacrylamide gel, which was fabricated by stacking and running gels, respectively, and subsequently subjected to electrophoresis with a current of 120 V in a mini-protein unit. The bands of protein in gels were stained with Coomassive brilliant blue R-250 (0.125%, *w*/*v*) for 4 h and then de-stained.

#### 2.3.9. Statistical Study of Films

All trials were studied at least in triplicate and ANOVA was used for all statistical variables. Duncan multiple range test (DMRT) was used to compare the (*p* < 0.05) difference between various concentrations of GTE at acidic and at alkaline pH. The significant difference was fixed at the level of 0.05. SAS Pro 8 (SAS Institute Inc., Carry, NC, USA) software was used to complete all statistical analyses.

## 3. Results and Discussions

### 3.1. Film Thickness

The thickness of film samples incorporated with distinct concentrations of GTE either at acidic (3) or alkaline (11) pH is shown in Table 1. All edible film thicknesses were in the range of 0.087–0.107 mm. Films containing plant extract at different concentrations had higher thicknesses compared to control films, which is in agreement with the results of Munir et al. [6] and Zhang et al. [24]. The GTE-incorporated film thicknesses were enhanced with the addition of extract and improved remarkably (*p*
**<** 0.05) with a 6% concentration at an alkaline pH. The thickness of GTE containing films may be affected by the addition of solid content and these solid contents can also increase the viscosity of edible films and improve the thickness [8]. Moreover, interactions between the hydroxyl groups of GTE phenolic compounds and surimi protein contributed to the film thickness. On the other hand, films incorporated with GTE at pH 11 had higher thicknesses as compared to pH 3. Different pH values and levels of GTE had a remarkable effect on the thickness (*p*
**<** 0.05). Zhang et al. [24] also described the addition of natural plant-based extracts having an effect on the thickness of protein-based films.

### 3.2. Film Solubility

The solubility of surimi films incorporated with different levels of GTE is presented in Table 1. Films incorporated with GTE had a relatively lower water solubility compared to the control. The solubility of films incorporated with GTE decreased with increasing levels of GTE at acidic and alkaline pH. Excessive solubility of control films demonstrated an additional number of hydrophilic compounds [25]. We assume that the inferior solubility of films incorporated with GTE extract could be accredited to the hydrophobic interactions of the phenolic compounds present in plant extracts [11]. Additionally, the water solubility of films was significantly lower in GTE-containing films at pH 3 compared to pH 11. The solubility of surimi films altered depending on the level of phenolic compounds in the GTE and the pH, correspondingly. Zhang et al. [24] also investigated that the FS of gelatin films with rosemary acid was lower as compared to control films. It was also posited that protein-based films are normally stabilized by various chemical interactions, such as disulfide, covalent, and noncovalent interactions, between molecules of plant extract (phenolic compounds) and proteins [11,25]. Therefore, the interactions between protein and phenolic compounds of GTE could be favorable for the formation of denser film structures, which could result in the lower attraction of water. However, Kaewprachu et al. [8] demonstrated contrary results in fish protein-based films incorporated with plant extracts, which could be attributed to the difference in their phenolic source, the type of protein, concentration of phenolic compounds, and pH.

### 3.3. Transparency

The transparency of surimi-based films incorporated with GTE at both pH 3 and 11 is illustrated in Table 1. Transparency of GTE-containing films either at pH 3 or pH 11 was much lower as compared to the control films and decreased as the concentration of GTE increased at pH 3. On the other hand, transparency at pH 11 decreased with 2% GTE and then increased as the concentration was increased. These results demonstrate that the transparency improved more significantly at pH 3 compared to pH 11 at higher concentrations. Moreover, the transparency of films was also affected by the pH and levels of plant extract, which might be due to the difference in concentration of their phenolic compounds and the type of interactions in different conditions [11,24]. Similar results were reported by Kaewprachu et al. [3] and Munir et al. [6] in fish-based protein film incorporated with plant extracts, but Arfat et al. [25] reported contrary results in fish skin gelatin and fish protein isolate blend films. Light transmission might be affected by variation in phenolic compounds in the film matrix as well as their bonding with protein molecules.

The protein films had excellent barrier properties for UV light owing to their high content of aromatic amino acids, which can absorb UV light [26]. The decline in light transmission is advantageous for food protection and preservation because UV light has been recognized to induce adverse effects commonly associated with lipid oxidation. The reduction in transparency might be due to the presence of coloring pigments in the GTE extract, which play an important role in decreasing the transmission of light. The results suggest that all phenolic compounds of GTE efficiently prevented the transmission of UV and visible light of protein films.

### 3.4. Water Vapor Permeability (WVP)

WVP is an important parameter related to vapor diffusion or transmission through films. The effect of pH and GTE on surimi-based edible films is presented in Table 2. The WVP of GTE-enriched films either at pH 3 or pH 11 was reduced with the increasing concentration of extracts, as compared to control films, and the lowest WVP was obtained at higher concentrations of extract. Films prepared with 6% GTE extract at pH 3 had the highest WVP out of all of the incorporated films, either at pH 3 or pH 11. These results indicate that the addition of GTE at different pH values, especially at higher concentrations, could change the surimi-based edible film WVP. Protein molecules in film-forming solutions, at pH 3 and at pH 11, were partially unfolded as a result of the denaturation of proteins and exposure of hydrophobic groups [26]. Kaewprachu et al. [8] and Ozdal et al. [27] also reported natural cross-linking among surimi proteins and phenolic compounds across hydrogen or hydrophobic bonds. These interactions are associated with a higher level of hydroxyl groups because a higher number of hydroxyl groups is indicative of a strong cross-linking between protein and phenolic compounds [11]. Furthermore, this was confirmed by reducing the intramolecular spaces of the protein polymer, as evidenced by a decrease in WVP [28]. Thus, the incorporation of GTE at different pH values is useful for improving the water diffusion properties by reducing the vapor transfer around the film atmosphere and the food. These outcomes are similar to those of earlier studies on fish protein-based films incorporated with plant extracts [8]. Friesen et al. [29] reported contrary results in soy protein-based films incorporated with epicatechin, which could be attributed to the difference in their phenolic source, the type of protein, the concentration of phenolic compounds, and pH.

The casting temperature of surimi proteins is very important in structural changes, as we casted our surimi films at 37 °C and the gelling point and amorphous phase of silver carp surimi proteins was above 37 °C according to the previous studies [30,31]. Furthermore, DSC thermograms were used to study the glass transition temperature (T_g_) of surimi-based edible films incorporated with green tea extract (as shown in Table 3), and a temperature range between −20 and 200 °C was selected. The DSC results were also confirmed by the mechanical properties of edible films that the TS decreased and the elongation at break was increased, as was also stated by other studies [32]. On the other hand, the authors in [31] found that palm oil in gelatin films can reduce the glass transition temperature (T_g_) in palm oil containing films compared to control. The study proved that GTE extract has a significant effect on the thermal properties of surimi protein-based edible films. On the other hand, we used plant-based extracts that were able to interact with proteins and yield gel with improved properties. This finding was similar to the investigation of Zhao and Sun [33], who acknowledged that cross-linked gelatin–polyphenol had markedly enhanced gel strength with a more compact surface.

In general, when phenolic compounds are incorporated into fish meat-based products, these compounds are oxidized and subsequently produce quinone species. As a result, the reaction of amino acid groups and quinones leads to enhanced gelling properties in which an elastic gel network can be formed [34]. Cao et al. [35] studied the impact of gallic acid at various concentrations on the gelling properties of oxidized myofibrillar proteins in which during the reaction, disulfide bonds were dominant. However, Tang et al. [36] also found that rosmarinic acid–protein had the adverse impacts on myofibrillar protein gelation. As a consequence, water holding capacity was decreased and gel strength was weakened [36].

Glass transition temperature (T_g_) of surimi-based edible films incorporated with green tea extract and a temperature range between −20 and 200 °C was selected. The DSC results were also confirmed by the mechanical properties of edible films that the TS decreased and the elongation at break was increased, as was also stated by other studies. On the other hand, Tongnuanchan et al. [37] found that palm oil in gelatin films can reduce the glass transition temperature (T_g_) in palm oil containing films compared to control. The study proved GTE extract has a significant effect on thermal properties of surimi protein-based edible films.

### 3.5. Mechanical Properties

Table 3 demonstrates the mechanical behavior of surimi films (TS and EAB). Surimi films incorporated with GTE either at pH 3 or pH 11 exhibited superior TS but inferior EAB, as compared to films without GTE. The TS of the GTE-incorporated films at pH 11 was higher as compared to the films incorporated with GTE at pH 3. Moreover, the TS increased as the concentration of GTE increased, and films incorporated with 6% GTE at pH 11 had the highest TS as compared to all other incorporated films. The increase in rigidity of films incorporated with GTE was probably due to cross-linking of proteins and phenolic compounds, including noncovalent and covalent bonding as well as disulfide or non-disulfide bonding [10,11]. Additionally, the interaction between phenolic compounds and proteins might decrease the effect of plasticizer in fish protein-based films and subsequently lead to superior stiffness of films [28]. These interactions contributed to a more compact structure and thereby improved the TS of surimi-based films. Likewise, some other studies reported an increased TS after the addition of plant extracts in fish protein-based films [6,8].

EAB percentage distinctly decreased with the accumulation of GTE either at acidic or alkaline pH. Films incorporated with 2% GTE at pH 11 had a higher EAB as compared to all the films incorporated with GTE at pH 3 or pH 11, indicating that the change in mechanical properties of films was highly affected by pH and the level of GTE. An increase in EAB was reported by Rattaya et al. [38] in fish skin gelatin film with seaweed extract, and a decrease in EAB was also noted by Kaewprachu et al. [8] in fish myofibrillar protein film incorporated with natural extract. Consequently, a change in the film’s EAB might be related to the difference in their protein and phenolic source, the concentration of phenolic compounds, and pH.

A hot air dryer (37 °C for 12 h) was used for drying the surimi films. After that, films were manually peeled off by hand and conditioned for four days in a conservational chamber to attain corresponding relative humidity (R.H 55–60%) of the final films for further analysis.

On the other hand, the moisture content of each film from five different random locations was measured, as presented in Figure 1. The moisture content of the prepared films was affected by concentration and pH and gradually decreased as the concentration of GTE increased, which might be due to the cross-linking between surimi protein and GTE phenolic compounds. Additionally, the reduction in hydrophilic groups also inhibit the water interaction in incorporated films. As a result, the moisture content of incorporated films decreased.

Additionally, plant-based polyphenols are able to interact with proteins, by covalent and non-covalent interactions, and form more rigid and thermally stable structures compared to pristine proteins [39]. Strauss and Gibson [40] found that mechanical strength and rigidity of gelatin gel incorporated with polyphenols were significantly increased. This conclusion was stated with Zhao and Sun [33], who acknowledged that cross-linked gelatin–polyphenol had markedly enhanced gel strength with a more compact surface.

The interaction between phenolic compounds and proteins is not only actually affected by phenolic compound structure but also by surface properties of proteins. Different proteins have different amino acid compositions, isoelectric points, and hydrophobic interactions, which impact the cross-linking ability of proteins with phenolic compounds [27].

### 3.6. Color

Table 4 shows the color of surimi-based edible films incorporated with GTE either at acidic or alkaline pH. In the determination of color, L*-value (lightness), a*-value (redness/greenness), b*-value (yellowness), and ΔE* were the main parameters to evaluate the color of films. Films incorporated with GTE had the lowest L*-value, which decreased as the concentration of GTE increased irrespective of pH. Control films showed an increased L*-value and an increased a*-value, and the lowest b*-value and the lowest ΔE* were found in films at pH 3 and pH 11, respectively. An inferior a*-value was noticed in films incorporated with GTE at pH 3 as compared to pH 11. On the other hand, films incorporated with GTE at pH 11 had a higher a*-value as compared to control films. The highest b*-values and ΔE* were obtained from films incorporated with GTE at pH 11, as compared to control. Incorporation of GTE in films improved b*-values, and ΔE* was also improved as the concentration of GTE increased at acidic as well as alkaline pH, while pH 11 produced higher b*-values and ΔE* color difference compared to pH 3. It was noticeable that the reduction in L*-values coincided with increases in b*-values and ΔE*, which were attained as a higher concentration of plant extract was incorporated, regardless of pH. The increase in b*-value of films with a higher level of GTE might be correlated with the higher concentration of natural color pigments in GTE, which played an important function in reducing the passage of light (Table 4). In addition, Kaewprachu et al. [8], and Prodpran et al. [41] reported similar results with regard to fish myofibrillar protein incorporated with plant natural extracts, but Arfat et al. [25] obtained contrary results from films based on fish skin gelatin. The color parameters of surimi-based films were influenced by the protein and phenolic source, the concentration of phenolic compounds, and pH.

### 3.7. Thermo-Gravimetric Analysis

The thermal stability of incorporated films with different levels of GTE either in acidic (pH 3) or alkaline (pH 11) conditions is presented in Figure 2A,B. Thermal degradation of films varied with pH and the concentration of green tea extract. The degradation temperatures (T_d_), weight loss (Δw), and residue of all surimi-based film samples are shown in Table 5. The results show that the films had four main phases of weight loss.

Phenolic compounds have abundant hydroxyl groups, which are important to mark the effect of cross-linking between phenolic compounds and proteins [11,27]. Additionally, the inter- and intra-molecular interactions, such as ionic, hydrogen covalent, and non-covalent conjugation, could also mark significant associations between phenolic compounds and proteins and might be due to the presence of hydroxyl groups and carboxyl groups [11,27], as illustrated in Figure 3. Additionally, enhanced thermal stability is also very important in the food packaging industry.

The initial phase of weight loss for all films (Δw_1_ = 3.8%–11.9%) was observed between the temperature (T_d1_) 127.7 and 177.6 °C. The loss of weight in this phase was possibly correlated with the loss of free and bound water absorbed in the films. Similar results were described in gelatin and red tilapia-based films [28].

Additionally, GTE incorporation at acidic and alkaline pH had greater weight loss as compared to the control films, and in this phase, weight was correlated with the sum of water in the film matrix. Films integrated with GTE at pH 11 had a higher weight loss in this phase as compared to pH 3.

The second phase of weight loss of all films was observed almost at the temperature of 187.6–275.4 °C (T_d2_) with Δw_2_ of 18.3%–26.5%. In this phase, the weight loss was mostly correlated with the loss of structural bound water and the low molecular weight of surimi protein glycerol [42]. It was perceived that Δw_2_ of the incorporated films with GTE at pH 3 was decreased and T_d2_ was increased; however, the T_d2_ and Δw_2_ of films incorporated with GTE at pH 11 were increased as compared to the control films. However, this range of temperature was higher than the boiling point of glycerol (182 °C), which may indicate that a specific type of cross-linking, such as hydrogen or covalent cross-linking, might exist between surimi proteins and glycerol molecules [11,42].

For the third phase of weight loss, the Δw_3_ of 10.1%–30.2% and the Td_3_ of 304.7–397.6 °C were observed for all films. This phase of weight loss was most probably correlated with the breakdown of the larger-sized interrelated proteins [42], and the Δw_3_ was related to the breakdown of dominant proteins in the film matrix. The T_d2_ and T_d3_ for myofibrillar protein films containing GTE were reduced as compared to those of the control films, although they were different with pH and concentration of plant extract, revealing that films incorporated with GTE presented greater heat resistance compared to control films.

Surimi-based edible films also had a fourth phase of weight loss. The fourth phase of weight loss (Δw_4_ = 4.1 – 20.1%) was observed at the temperature (T_d4_) range of 423.6–495.1 °C. Weight loss in this phase was mostly associated with the loss of high-temperature stable components. Furthermore, increased T_d4_ and decreased Δw_4_ of the films were observed in all films incorporated with extracts at either pH 3 or pH 11. The addition of GTE had a remarkable impact on the thermal stability of films due to the phenol–protein interaction (*p* < 0.05), especially by covalent cross-linking [11]. Films incorporated with 6% GTE at pH 3 demonstrated higher residues than all other films incorporated with GTE either at pH 3 or pH 11.

### 3.8. Protein Pattern

The protein pattern of surimi films incorporated with GTE either at acidic or alkaline pH is presented in Figure 4. All band intensities were amplified as compared to control films. Films incorporated with GTE at different pH values had a myosin heavy chain (MHC) and a myosin light chain (MLC) as leading proteins, but films incorporated with GTE at pH 3 had a lower band intensity of MHC and MLC as compared to those incorporated at pH 11 (Figure 3). Furthermore, films incorporated with GTE at pH 11 had a marked difference in upper and lower bands. Polymerized MHC with a higher molecular weight (above 250 kDa) was formed in the films incorporated with GTE at pH 11, as evidenced by a band at the top of the SDS–PAGE gels. The band intensity was enhanced with the increasing concentration of GTE, which indicates a strong interaction between phenolic compounds of GTE and surimi proteins. Additionally, it was noticed that the change in protein pattern was strongly dependent on pH. The intensity of MHC was higher at pH 11 as compared to pH 3. The incorporation of GTE might modify the configuration of disulphide and non-disulfide bonds by covalent modification during protein–phenol interactions [2,10]. Additionally, the 250 kDa band (MHC) entirely disappeared in GTE-containing films, which might be due to the degradation of MHC caused by cysteine proteinase, endogenous cathepsin D [43], and heat treatment [44]. These results are in agreement with the mechanical, barrier, and thermal properties of films incorporated with GTE (Table 2).

## 4. Conclusions

The physico-mechanical, thermal, and barrier properties of films were maintained after the addition of GTE in surimi-based edible films either in acidic (pH 3) or alkaline (pH 11) conditions, as compared with the controls. The addition of GTE demonstrated a significantly greater influence on the film properties at pH 3 as compared to pH 11, which indicates that pH 3 is more suitable for cross-linking between proteins and phenolic compounds in all GTE-containing films. The concentrations of plant extract and pH had different effects on the properties of the films. The enhanced properties of the films were most likely caused by the appropriate cross-linking between protein and phenolic compounds in GTE at pH 3 or pH 11 (Figure 3).

Films incorporated with GTE at acidic or alkaline pH could be used as potential active food packaging materials to improve the quality and extend the shelf life of packed foods by improving the TS, the water vapor barrier, the transparency, the film solubility, the color, and the thermal stability.

## Figures and Tables

**Figure 1 polymers-12-02281-f001:**
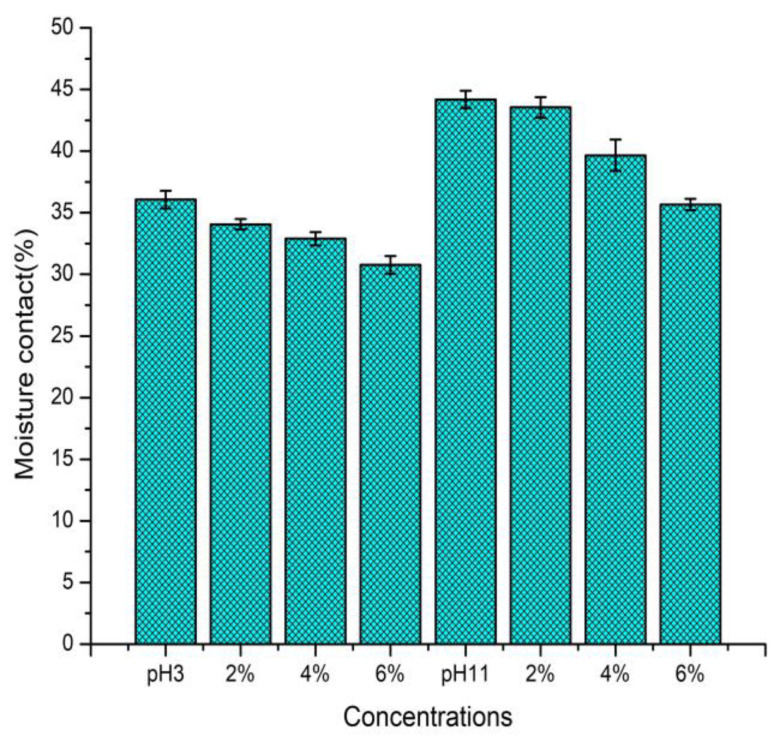
Moisture content of each film from five different random locations was measured.

**Figure 2 polymers-12-02281-f002:**
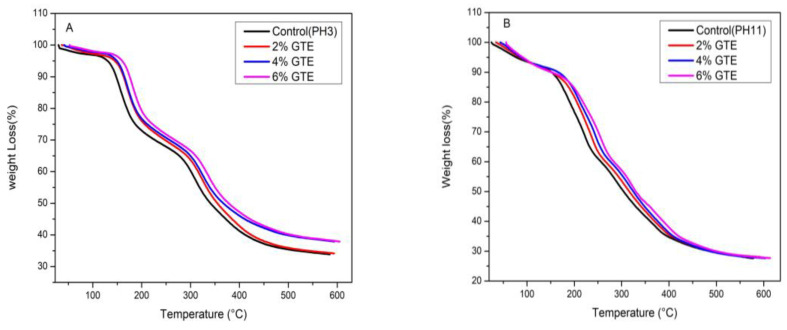
(**A**,**B**). Thermo-gravimetric curves of weight loss for silver carp surimi-based edible films at pH 3 and pH 11 incorporated with different concentrations of green tea extract (GTE). (**A**) GTE (pH 3) (2%, 4% and 6%); (**B**) 2%, 4%, 6% GTE pH 11, Control (without added GTE at both pH).

**Figure 3 polymers-12-02281-f003:**
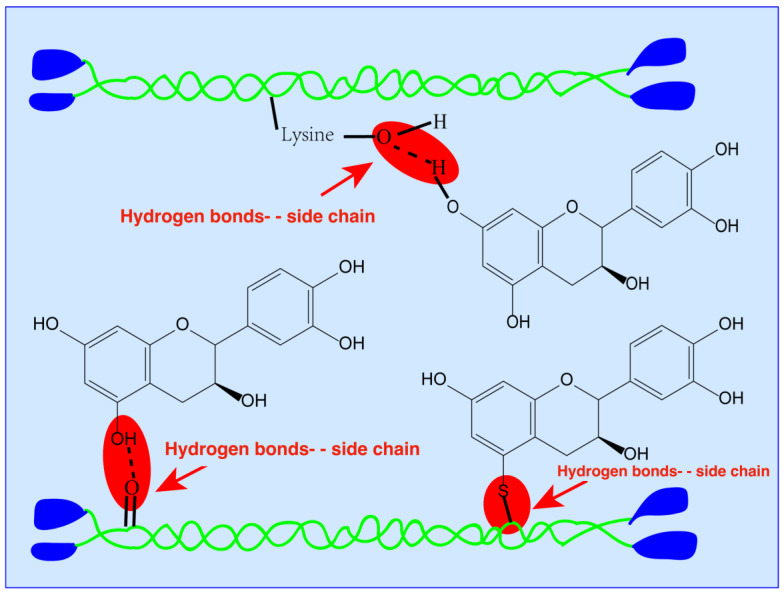
Graphical illustration of cross-linking mechanism between GTE and surimi proteins.

**Figure 4 polymers-12-02281-f004:**
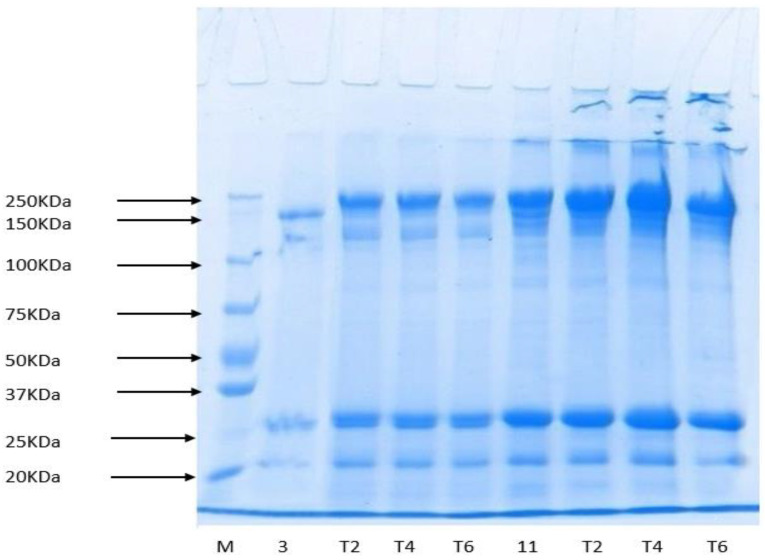
Protein patterns of surimi-based edible films containing different concentrations of green tea extract (GTE) at pH 3 and pH 11. M: Marker, 3: Control with pH 3 and 11: Control with pH 11 (a) T2, T4, and T6: 2%, 4%, and 6% GTE at pH 3 and pH 11.

**Table 1 polymers-12-02281-t001:** Thickness, film solubility (FS), and transparency of films incorporated with green tea extract (GTE) at pH 3 or pH 11 with different concentrations.

PH Level	Source	(*w*/*w* Protein)	Thickness (mm)	FS (%)	Transparency (% mm^−1^)
pH 3	Control	Control	0.087 ± 0.002^F^	54.76 ± 0.17^B^	3.11 ± 0.17^C^
	GTE	2%	0.088 ± 0.002^F^	36.90 ± 0.13^E^	3.28 ± 0.06^CB^
		4%	0.093 ± 0.001^E^	32.95 ± 0.17^G^	3.38 ± 0.20^B^
		6%	0.098 ± 0.002^D^	30.48 ± 0.2^H^	3.59 ± 0.18^A^
pH 11	Control	Control	0.92 ± 0.002^E^	64.22 ± 0.08^A^	0.96 ± 0.08^F^
	GTE	2%	0.100 ± 0.002^C^	42.65 ± 0.09^C^	3.11 ± 0.15^C^
		4%	0.103 ± 0.002^B^	39.66 ± 0.14^D^	2.38 ± 0.16^D^
		6%	0.107 ± 0.002^A^	36.27 ± 0.17^F^	1.96 ± 0.18^E^

Data are presented as means ± SD for thickness, FS, and transparency (n = 6). Different letters in the same column indicate significant differences (*p* < 0.05) among the film samples with different concentrations of GTE at different pH levels.

**Table 2 polymers-12-02281-t002:** Water vapor permeability (WVP), tensile strength (TS), and elongation at break (EAB) of films incorporated with GTE at pH 3 or pH 11 with different concentrations.

PH Level	Source	(*w*/*w* Protein)	WVP (×10^−12^ gm^−1^ S^−1^ pa^−1^)	TS(MPa)	EAB (%)
pH 3	Control	Control	3.15 ± 0.04^B^	2.57 ± 0.14^G^	164.77 ± 3.06^B^
	GTE	2%	2.40 ± 0.03^ED^	3.97 ± 0.20^E^	88.95 ± 6.62^F^
		4%	2.22 ± 0.04^F^	4.17 ± 0.10^D^	97.06 ± 5.80^F^
		6%	2.06 ± 0.05^G^	5.66 ± 0.17^A^	131.38 ± 5.32^E^
pH 11	Control	Control	3.63 ± 0.05^A^	2.74 ± 0.09^F^	187.35 ± 5.94^A^
	GTE	2%	2.68 ± 0.09^C^	5.18 ± 0.11^C^	154.21 ± 8.68^C^
		4%	2.46 ± 0.04^D^	5.45 ± 0.06^B^	144.20 ± 10.55^D^
		6%	2.32 ± 0.08^EF^	5.68 ± 0.08^A^	122.26 ± 13.27^E^

Data are presented as means ± SD for WVP (n = 3) (TS and EAB n = 6). Different letters in the same column indicate significant differences (*p* < 0.05) among the film samples with different concentrations of GTE at different pH levels.

**Table 3 polymers-12-02281-t003:** Thermal transition temperature of surimi-based edible films incorporated with green tea extract.

pH	Sample	Thermal Transition Temperature (T_g_)
pH3	Control	59.5
	2%	91.1
	4%	115.4
	6%	128.9
pH11	Control	65.2
	2%	97.7
	4%	108.1
	6%	133.8

**Table 4 polymers-12-02281-t004:** The color determination of surimi-based edible films incorporated with different concentrations of GTE at pH 3 or pH 11.

PH level	Source	(*w*/*w* Protein)	L*	a*	b*	ΔE*
pH 3	Control	Control	91.49 ± 0.07^A^	–0.36 ± 0.03^C^	1.42 ± 0.06^G^	7.27 ± 0.11^G^
	GTE	2%	90.69 ± 0.05^B^	–0.35 ± 0.03^C^	2.30 ± 0.05^F^	9.29 ± 0.08^F^
		4%	89.40 ± 0.04^C^	–0.67 ± 0.04^D^	4.40 ± 0.03^E^	13.37 ± 0.08^E^
		6%	88.79 ± 0.07^D^	–0.95 ± 0.03^E^	7.48 ± 0.05^C^	18.11 ± 0.09^C^
pH 11	Control	Control	91.53 ± 0.04^A^	0.22 ± 0.03^B^	1.25 ± 0.06^H^	7.15 ± 0.08^H^
	GTE	2%	86.66 ± 0.14^E^	0.21 ± 0.03^B^	4.76 ± 0.16^D^	17.69 ± 0.12^D^
		4%	83.50 ± 0.12^F^	0.21 ± 0.02^B^	12.60 ± 0.12^B^	31.08 ± 0.18^B^
		6%	81.75 ± 0.10^G^	0.13 ± 0.03^A^	20.72 ± 0.10^A^	44.51 ± 0.16^A^

Data are presented as means ± SD (n = 9). Different letters in the same column indicate significant differences (*p* < 0.05) among the film samples with different concentrations of GTE at different pH levels.

**Table 5 polymers-12-02281-t005:** Thermal degradation temperature (Td, °C) and weight loss (Δw, %) of films incorporated with GTE at different concentrations of green tea extract (GTE) at pH 3 and pH 11. Δ_1_, Δ_2_, Δ_3_, and Δ_4_ represent first, second, third, and fourth stages, respectively, of weight loss of films during the heating scan.

Samples	(*w*/*w* of Protein)	Δ1		Δ2		Δ3		Δ4		Residue (%)
		Td1	Δw1	Td2	Δw2	Td3	Δw3	Td4	Δw4	
pH 3	Control	127.7	4.9	187.6	20.7	312.7	18.5	455.2	18.3	35.6
	2%	144.7	4.5	209.7	25.2	319.1	16.6	445.1	17.1	36.6
	4%	147.2	4.9	205.2	18.3	304.7	13.9	427.2	19.4	43.5
	6%	148.5	3.8	221.1	21.5	309.5	10.1	423.6	20.1	44.5
pH 11	Control	160.4	11.3	235.3	24.5	397.6	30.2	487.6	4.1	29.9
	2%	177.6	11.9	251.9	25.3	392.6	27.4	495.1	5.6	29.8
	4%	167.1	10.0	263.6	25.5	391.1	23.8	467.6	6.8	33.9
	6%	176.3	11.7	275.4	26.5	337.5	13.4	440.6	14.3	34.1

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
