# Peer review of "The Effect of Acidic and Alkaline pH on the Physico-Mechanical Properties of Surimi-Based Edible Films Incorporated with Green Tea Extract"

_polymers, 2020, doi:10.3390/polym12102281_

Round 1

Reviewer 1 Report

The manuscript deals with Physico-mechanical properties of surimi based edible films incorporated with green tea extract. The topic is of enough interest. The rersults are interesting, valuable and applicable. The introduction part is well written. All in all, the manuscript reaches appropriate parameters for journal Polymers.

However, there are present two points which decrease overall quality of the manuscript:

  1. Part 2.3.2: Supplier of the instrument is not Heidolth but most probably it is Heidolph. Moreover, the type of used instrument is most probably not Incubator 10000 but Incubator 1000. 
  2. The level of English is more less adequate. Nevertheless, it is absolutely inappropriate to mix together American English spelling eg. ("vapor" in part 3.4) and ("color" part 3.6) with English spelling which is prefered everywhere else as eg. ("behaviour" part 3.5). Thus, the spelling has to be unified. I would accept American English spelling as well as spelling used everywhere else but never their mixture which is absolutely inappropriate. Therefore, the English has to be thoroughly checked and manuscript has to be adequately corrected and rewritten.  

Author Response

Find attached file

Reviewer 2 Report

Major revision are necessary

-References 9, 12 and 23 are present in the list of references but are not reported in the text

-pag.2 line78 ..’2.5% NaCl (weight of surimi)..’ it is not clear it means w/w of surimi?

-pag.2 line 80-81 ...’60%(w/w)of protein’...but there is not method to measure the protein concentration

-pag.4 line 143 ...’15ml of each film prepared sample was loaded onto the polyacrylamide gel.’...The amount loaded onto the gel is between 10-100µl max

-pag.5 line167 ...the water solubility of films (WS) in the table 1 is called film solubility (FS). Also in the methods (pag.3 line 97)  it is not clear: the short name should be WS or FS.

-pag7 line 261..’fig.1 (A,B)’ are not present in the text nor in the supplementary material, so is impossible to understand the comment

-pag.7 line 296 ..’fig.2’ is not present in the text nor in the supplementary material, so is impossible to understand the comment

-pag.7 line 305 ...additionally the band of 250KDa (MHC) entirely disappeared.’...It disappeared because polymerized at higher molecular weight, it is scarcely likely to be degraded, in the experimental condition of the authors, in the ref. 33 the heat treatment play an important role.

The paper needs improvement. The role of the proteins and the interactions with the green tea should be studied in deep. The paragraph on electrophoresis must be improved.

Author Response

Find attached file

Reviewer 3 Report

Manuscript Draft ID polymers-911223

PHYSICO-MECHANICAL PROPERTIES OF SURIMI BASED EDIBLE FILMS INCORPORATED WITH GREEN TEA EXTRACT AS AFFECTED BY ACIDIC AND ALKALINE PH

Major revision for acceptance

This manuscript describes a study focused in understanding the effect of green tea extract at different pH (acidic or alkaline), on mechanical, thermal and permeability of edible films based on silver carp surimi.

The introduction section requires an improvement in English language, but also a better explanation about the relevance of this study. From the first paragraph is not seem clear why is important produce biomaterials from silver carp for active packaging. Authors also need to better state the bioactive properties of tea extracts (antioxidants, antimicrobial), in this sense the manuscript lacks and adequate literature review. The relevance to study the role of pH is not evident from previous sections of the manuscript. Therefore, the main objective needs to be rewritten, indeed the last sentence in this section doesn’t make sense with the rest of text.

Regarding with the methodology, in section 2.2. is not clear why the authors have to prepare an emulsion from the raw surimi. In section 2.3.2 authors have to better explain that “pre-dried surimi films” are containing tea extract. Maybe an acronym may be helpful? Section 2.3.4. also must be better explained, is not clear the protocol used for this section. In section 2.3.5. authors need to include the sample thickness, because the mechanical response will be dependent of the sample thickness. Section 2.3.7. also needs more details in protocol: temperature range tested, heat rate, amount of sample, etc.  

Results and Discussion

Section 3.1: Regarding with thickness results, authors could explain why do they mention about the thickness is “enhanced” or “improved” by the addition of tea extracts? This section must be rewritten too. Main ideas are continuously repeated across the text without adding more inputs to the whole discussion (e.g. second half of paragraph is basically the main idea discussed on the first part of this section).

Section 3.2: authors must decline to use the term “water solubility” (as is used in this section) by “films solubility” (as is declared in Table 1). Also, they should change the term “incorporated film”, is confuse. The authors have suggested some explanation for the results obtained, however would be recommended contrast their results with similar protein films systems. Literature is generous in this sort of studies.

Section 3.3: normally optical properties in polymer films are also explained in terms of the Mie or Rayleight scattering. This should be also considered by the authors in this section. On the other hand, looking more carefully in the methodology, is it no clear how the authors have assessed the transparency in %. If the authors have considered the film thickness, so the values reported in Table 1 should be expressed in % mm-1. Moreover, the potential effect of isoelectric point of myofibrillar proteins haven’t been considered by the authors. Do you have an idea about the PI of these proteins? The structural configuration of proteins is dependent of the PI and could help to explain results obtained in terms of solubility and transparency.

Section 3.4: regarding with permeability, in this point the PI also have a relevant role. Permeability normally is discussed in terms of the structural features, therefore how the protein folded/unfolded and exposing their polar groups to interact with the tea extracts, will determine how the water molecules can transport across the film. However, also the structure is strongly defined by the casting temperature, in this work done at 37ºC. What is the gelling point of these proteins? This is a key point because will define if the system is semi-crystalline or amorphous, and these structures have different behavior in terms of transport properties.

Section 3.5: in line with previous section, mechanical properties of these films are strongly influenced by structural features. Authors should mention the moisture content of each sample, in order to clarify if the results are influenced by the presence of water. Along with PI and the casting temperature, the glass transition temperature could be also giving information about the structure of these films. Results reported by the authors are not only explained by a crosslinking effect protein-phenol.

Section 3.7: thermal stability is a very useful parameter, but authors should better explain the importance of this measurement to the main goal of the work. It looks a bit unconnected to the rest of the work. An image showing the weight loss as a function of temperature would complement the table data. Despite in the text a “Figure 1” is cited, it was not included embebed into the manuscript nor in supplementary files…

Section 3.8: the manuscript submitted didn’t included any figures, so this section of the draft is not possible to correct in the current way. Please correct this mistake and submit again.

Despite this manuscript doesn’t include all the elements to evaluate the work, it would be recommended to consider previous suggestions related with the inclusion of PI effect on the films properties and some other structural features (glass transition, moisture content, etc). Also the English language must be improved. Various sections of the manuscript lacks of clarity, therefore I would suggest to the authors to use a English correction service before to resubmit this manuscript. 

In Keywords sections please correct the “physio-mechanical” word. And “plant extracts” by “green tea extracts”.  

Author Response

Find attached file

Round 2

Reviewer 2 Report

I think the article improved with the latest changes.

Author Response

kindly find the attached file

thank you

Reviewer 3 Report

This reviewed draft shows some improvements respect to the first version submmited. However, important aspect have to be corrected before to be accepted for publication. 

For instance, between line 61 and 66 the authors should include a sentence in order to introduce the explanaition about the relevance of pH in this work. 

The rest of required corrections are related with aspects not solved by the authors in this reviewed version.

Section 3.1: Regarding with thickness results, authors could explain why do they mention about the thickness is “enhanced” or “improved” by the addition of tea extracts? This aspect remains unclear in this version.

Section 3.2: authors must decline to use the term “water solubility” (as is used in this section) by “films solubility”. Authors are testing how the film solubilizes in water, to be consistent with definition gave by Eq.1. Also, they should use the term “incorporated film” in the whole text. On the other hand, the authors have suggested some explanation for the results obtained, however would be recommended contrast their results with similar protein films systems. Literature is generous in this sort of studies. It should be seriously considered by the authors.

Section 3.3: normally optical properties in polymer films are also explained in terms of the Mie or Rayleight scattering. This should be also considered by the authors in this section. On the other hand, looking more carefully in the methodology, is it no clear how the authors have assessed the transparency in %. If the authors have considered the film thickness, so the values reported in Table 1 should be expressed in % mm-1. These two comments were not considered by the authors.

Section 3.4: regarding with permeability, the structure is strongly defined by the casting temperature, in this work done at 37ºC. What is the gelling point of these proteins? This is a key point because will define if the system is semi-crystalline or amorphous, and these structures have different behavior in terms of transport properties. This comment was not considered by the authors.

Section 3.5: in line with previous section, mechanical properties of these films are strongly influenced by structural features. Authors should mention the moisture content of each sample, in order to clarify if the results are influenced by the presence of water. Along with PI and the casting temperature, the glass transition temperature could be also giving information about the structure of these films. Results reported by the authors are not only explained by a crosslinking effect protein-phenol. This comment was not totally solved by the authors.

Section 3.7: thermal stability is a very useful parameter. Although  a plot with the behavior of samples during testing was included and look very nice, the authors should better explain the importance of this measurement to the main goal of the work. It remains a bit unconnected to the rest of the work. 

It would be strongly recommended to the authors using a english translator service to correct the next version in order to improve the english language. Text in some sections lack of clarity. 

Author Response

Kindly find the attached file

Thank you
